# Role of Comparative Advantage in Biofuel Policy Adoption in Latin America

**Ram N. Acharya [1],\* and Rafael Perez-Pena [2]**

1   Department of Agricultural Economics and Agricultural Business, New Mexico State University, Las Cruces, NM 88003, USA
2   Hunt Institute for Global Competitiveness, College of Business Administration, University of Texas at El Paso, El Paso, TX 79902, USA; rperezpena@utep.edu
\*   Correspondence: acharyar@nmsu.edu; Tel.: +1-575-646-2524

**Abstract:** The primary objective of this study is to evaluate whether renewable energy initiatives recently developed and implemented in Latin American and Caribbean countries are consistent with their national resource endowments, policy goals, and the general postulates of economic theory. Most classical and neoclassical theories suggest that international trade enhances economic efficiency and welfare of both parties involved in the exchange when they focus on producing and distributing products and services in which they have a comparative advantage. To achieve this goal, we analyze ethanol policy drivers using panel data from four major economies—Argentina, Brazil, Colombia, and Mexico. Since there is no universally accepted measure of comparative advantage, three separate models with different indicators—relative feedstock price, comparative export performance, and revealed comparative advantage—along with control variables, including the availability of production resources such as land and farm labor, are estimated. As expected, results show that the comparative advantage in feedstock production was one of the crucial factors in determining biofuel policy development and implementation in the four countries.

**Keywords:** comparative advantages; biofuel policy; ethanol; Latin American countries

---

## 1. Introduction

Global concerns about rising fossil fuel consumption and its likely impact on climate, human health, and the overall economy have spurred interest in developing and implementing renewable energy policies throughout the world. Most of these policies envision domestic production and utilization of first-generation biofuels such as ethanol and biodiesel [1,2]. Since corn, soybeans, and sugarcane are primary biofuel feedstocks, these energy policy initiatives compete directly with the human food supply system [3–5]. Therefore, it is crucial to understand whether renewable energy initiatives are consistent with domestic production capabilities and resource endowments (e.g., land, labor, and technology) and how they impact other sectors of the economy [1,3,5–7].

Moreover, most renewable energy initiatives have multiple objectives, including the reduction of greenhouse gas ($CO_2$) emissions [2,5], boosting farm incomes [1,8], enhancing agricultural development [9,10], reducing dependence on imported oil [11,12], and promoting overall economic and environmental sustainability [3,7,13,14]. However, it is not clear whether such multifaceted policy goals are consistent with economic theory in general and international trade theory in particular [1,9,15–21].

Economic theory postulates that international trade promotes efficiency by allowing countries to specialize in producing goods and services in which they are most proficient—i.e., have a comparative advantage in the global market [22,23]. Ricardian, as well as Heckscher–Ohlin models, are widely used to identify sources of comparative advantage and measure its impact

on trade and other economic sectors [17,24,25]. While the classical theory assumes that the relative advantage arises because of differences in factor productivity (e.g., labor and land) between trading partners [20], the Heckscher–Ohlin factor-proportions theory (neoclassical trade theory) focuses on factor endowments [15]. The neoclassical theory views market goods as packets of different input combinations used in producing them, such as labor, land, and capital. In other words, all traded products reflect factor endowments of the trading partners [24]. In this regard, international trade is a vehicle for transferring services of immobile production resources like land from locations where they are in abundance to places where they are in short supply [15,24,26,27].

Despite these fundamental differences, the classical and neoclassical theories both postulate that trade provides an opportunity to increase production efficiency and enhance the welfare of both parties involved in the exchange [28–30]. Since trade is a significant activity of most economies, not only the availability of essential production resources such as land and labor but also the comparative advantage in producing a specific crop, such as sugarcane that is used as a feedstock, is likely to play a crucial role in determining the long-term economic viability and sustainability of biofuel policies recently adopted in four Latin American and Caribbean (LAC4) countries (Argentina, Brazil, Colombia, and Mexico).

The whole Caribbean and the Latin American region are classified as high potential areas for producing energy crops [19,31]. Although sufficient supplies of fertile land and labor are essential for producing biofuel feedstock, they may not be enough to gain a competitive advantage in the global market. Comparative advantage depends more on the capabilities of a region or country to produce and supply the traded product at the lowest cost consistently than on resource availability. Consequently, a better measure is required to gauge the actual impact of comparative advantage on biofuel policy adoption and implementation.

In search of a better measure, a wide range of proxies based on either resource availability (e.g., arable land and other natural resources) or market prices and trade patterns (e.g., observed market prices and international trade data) have been proposed and used in the literature [17,32–38]. We argue that access to production resources, such as land, labor, capital, and other natural resources, is crucial for producing biofuel feedstock. Feedstock prices and international trade patterns are used to construct the comparative advantage indices, using an approach consistent with the existing literature [32,33]. These comparative advantage indices have not yet been widely used for biofuel policy analysis. In this light, the current study contributes to the existing literature by evaluating the impact of three different comparative advantage measures (i.e., relative feedstock price, relative export performance, and revealed comparative advantage) along with other crucial policy variables including technology, rural development, energy independence, and pollution control on biofuel policy adoption in the LAC4 countries.

As expected, our empirical results show the probability of a country adopting biofuel policies rises as per capita gross domestic product (income), per capita arable land (land), access to modern technology (proportion of population using the internet), and availability of farm labor (the percentage of economically active population in agriculture) increase. Likewise, countries heavily dependent on imported oil are more likely to adopt biofuel policies. The results also show that countries with a higher proportion of $CO_2$ emissions from liquid fuels are less likely to adopt biofuel policies. Moreover, as postulated by international trade theories, comparative advantage in feedstock production played a crucial role in biofuel policy development and adoption in the LAC4 countries.

The remaining segments of the paper are organized as follows: Section 2 reviews renewable energy policies in leading biofuel producing and the LAC4 countries; Section 3 presents the conceptual framework and estimation models based on existing literature, defines model variables, describes data sources, and defines the expected relationships between the model variables; the fourth section presents and interprets the empirical results; and the final section draws conclusions and explains some of the limitations of the study.

## 2. Review of Energy Policies in Leading Biofuel Producing and LAC4 Countries

Global concerns about rising fossil fuel consumption and its potential impact on the ozone layer, air pollution, human health, and climate change have encouraged both developed and developing countries to consider sustainable and cleaner-burning fuel alternatives, such as biofuels [39–44]. As a result, world production of bioethanol increased more than fivefold from about 17 billion liters in 2000 to 108 billion liters in 2018 (for details see the annual biofuel production data posted on the Renewable Energy Association webpage: https://ethanolrfa.org/). However, both production and utilization are highly concentrated in only a few regions. The top three ethanol producers—the United States (56 percent), Brazil (28 percent), and the European Union (5 percent)—contributed nearly ninety percent of the total 2018 global supply. Although the total number of countries with formal biofuel policies have reached sixty-five by 2018, the aggregate contribution of the remaining sixty-two countries that includes China (four percent), Canada (two percent), India (one percent), Thailand (one percent), and Argentina (one percent) in the global market is about ten percent.

The most effective policy tools for implementing the biofuel policies include biofuel–fossil fuel blending requirements, tax subsidies, and flex-fuel vehicle programs [40,41,43,44]. Although a wide range of feedstocks are used for producing the first-generation biofuels, corn (USA) and sugarcane (Brazil) are the primary sources for making ethanol and oilseeds (e.g., soybeans, palm oil, rapeseed, castor beans, and jatropha) are used for producing biodiesel. Among bioethanol feedstocks that are currently in use, the yield for sugarcane (5472 L/hectare) is much higher than for corn (3751 L/hectare). For biodiesel, the yield from oil palm (4733 L/hectare) is much higher than from jatropha (1590 L/hectare) and castor beans (1310 L/hectare). Next-generation biofuels are likely to come from non-edible plant biomass (e.g., cellulosic switchgrass) and microalgae that have yield potentials of 10,757 and 46,957 L/hectare, respectively. A short description of policies in the significant biofuel producing countries, along with their program objectives and feedstock use, is provided below.

The U.S.: Concerns for air pollution, climate change, energy security, rural development, and the increasing trade imbalance because of the heavy dependence on imported fossil fuel have been the primary driving forces at different times in advancing biofuel policy in the U.S. over time [12,45,46]. In particular, the Clean Air Act of 1963 and its subsequent amendments and added policy-driven uses of reformulated gasoline to reduce the emission of ozone and toxic air pollutants from petroleum products were the main driving forces in accelerating the production and utilization of biofuel products in the U.S. Energy Independence and Security Act of 2007 set a goal of reaching an eighteen percent renewable fuel use in the transportation sector by 2022. The Biomass Program in 2008 established an additional target of reducing gasoline consumption by seventy percent by 2030. The blending requirements and a biofuel production subsidy have been the primary policy tools used in promoting the industry. Because of the relative advantage in growing and processing corn and its potential for supporting the rural economies, corn became and remained the primary feedstock for producing bioethanol in the U.S. Although it may not be the best feedstock for creating bioethanol, corn played a critical role in making the U.S. one of the leading producers, consumer, and exporters of biofuels in the global market.

Brazil: The increasing desire for energy security (mainly after the energy crisis of 1973), reducing the heavy dependence on imported oil, and enhancing economic development were the primary motivations for implementing various biofuel policies in Brazil [9,47]. After the implementation of the National Fuel Alcohol Program in 1975, Brazil became the global leader in biofuel production and utilization until 2005, when the U.S. surpassed it on both accounts. In particular, it has been highly successful in utilizing several policy tools, including ethanol–petrol blending requirements (20%–30%), tax subsidies (e.g., reduction in value-added and fuel taxes), and the flex-fuel vehicles program to promote the biofuel industry [9]. Brazil uses sugarcane as the primary feedstock for producing bioethanol primarily because of its higher productivity and availability of abundant fertile land resources. In its recent effort to revitalize the industry, the new National Biofuel Policy "RenovaBio" aims to reduce greenhouse gas emission by 37% and 43% by 2025 and 2030, respectively, from its

baseline emission level of 2005. The new policy also proposes to develop a mechanism to privatize the decarbonization credit allocation process [47].

EU: Although biofuels were widely used in several European countries (e.g., France and Germany) before and during World War I, the oil crisis of 1973 provided the impetus for revitalizing the industry for enhancing energy security and rural development [48]. However, large-scale biofuel production started only after the 2003 Biofuel Directive established the biofuel–fossil fuel blending target of 7.5%. Moreover, a 2009 EU Directive set two separate goals of biofuel utilization: a) to increase the share of renewable fuel to twenty percent in gross domestic energy consumption and b) to increase the transportation sector's use of renewable fuel to ten percent by 2020. In 2011, the EU set additional targets for reducing greenhouse gas emissions, forty percent by 2030, sixty percent by 2040, and eighty percent by 2050, respectively. A combination of tools, including a blending requirement, tax incentives, and emission targets are used to promote the policy goals that have led to increasing the use of biofuels in the transportation sector from about one percent in 2004 to nearly six percent in 2014.

Argentina, Colombia, and Mexico also joined the global renewable energy revolution by enacting various biofuel laws and promoting the production and consumption of biofuels [41,49,50]. Although Argentina started producing bioethanol after the energy crisis of the 1970s to ensure energy security and promote economic development, the passage of a 2006 law promoting sustainable production and utilization of biofuels revitalized the dormant renewable energy sector making it one of the leading exporters of biodiesel in the global market by 2010 [41,49]. The policy instruments used for promoting biofuel production in Argentina included a mandatory five percent blending requirement, tax subsidies, and other provincial and national initiatives supporting feedstock production and biofuel processing enterprises [49].

Colombia passed a law (Law 963) in 2001 to promote domestic biofuel production and utilization that mandated reaching a ten percent bioethanol–fuel blending requirement by 2005 for large cities with more than 500,000 inhabitants. Subsequent legislation and regulations make provisions for production (Law 939 of 2004) of biodiesel and flex-fuel vehicles (Decree 1135 of 2009). The policy goals emphasized in these regulations include economic development (by supporting the sugarcane and palm oil sector), diversification of energy sources, and reduction in greenhouse gas emissions by utilizing environmentally friendly fuels. The choice of feedstock is sugarcane for producing ethanol and palm oil for biodiesel [50].

Compared to other countries of the LAC region, Mexico has been relatively slower in developing and implementing biofuel policies. One of the reasons for its cautious approach to biofuel production might be its sizeable state-owned petroleum sector. The fossil fuel industry, which is considered to be strategically important, generates about eight percent of the total export revenue, contributes more than thirty-five percent to the federal budget, and attracts more than fifty percent of the public investment [39]. Despite the massive investment in the fossil fuel industry, Mexico enacted a Bioenergy Promotion and Development Law in 2008 to diversify domestic energy production, revitalize the rural economy, control air pollution, and reduce greenhouse gas emissions [40,51]. Subsequently, several initiatives promoting local production of biofuels from cactus, castor beans, jatropha, soybeans, and palm oils were initiated. Finally, regulations requiring biofuel and fossil fuel blending were implemented in 2017, and after a lengthy court battle, Mexico began importing biofuels from different sources, including the U.S., to meet its blending requirements.

## 3. Research Methodology

### 3.1. Conceptual Framework and Models

Limited dependent variable models, such as logit or probit regressions, have been used in most empirical studies to analyze technology adoption decisions [52–55]. Consistent with this literature, a simple conceptual model of biofuel policy adoption decision can be specified as

$$Bpol = f(X_i) \tag{1}$$

where *Bpol* is a binary variable that takes the value of one if a specific country develops and adopts a biofuel policy in period t and zero otherwise. The explanatory variables, $X_i$s, include all potential drivers of biofuel policy development and implementation.

The adoption of renewable energy policy and how biofuels are produced in a particular country generally depends on various factors. Among these factors are the country's initial endowment of production resources (e.g., land, labor, and other resources), access to production technologies, technical know-how, and the extent of global competition [19,56]. For example, U.S. biofuel policy initiatives, in general, encourage corn-based ethanol production mainly because of their potential for expanding domestic corn production, easy access to corn-based ethanol production technologies, and the possibility of enhancing farm income [19,57,58]. On the other hand, Brazil focused on sugarcane-based ethanol production for similar reasons [3,19].

Since the primary objective of this study is to evaluate whether renewable energy initiatives of individual countries are consistent with the resource endowments, multiple policy goals as outlined, and the general postulates of the economic theory, Equation (1) is revised by including several relevant variables. As discussed earlier, availability and access to biofuel feedstock production resources, such as arable land (*Land*), the agricultural labor force (*Labor*), as well as technological know-how and access to efficient production technologies (*Tech*), are critical for producing biofuel products. Moreover, recent studies show that per capita income plays a crucial role in shaping the environmental policies of a country. In particular, the impact of income on renewable energy initiatives are expected to be nonlinear or inverted U-shaped [59]. Therefore, two-income terms, *GDP* and *GPD²* (i.e., *GDP* squared), are used to account for this relationship.

Moreover, most biofuel policy reports mention rural development, energy independence, and pollution control as other policy objectives [2,5]. Since countries with sizeable rural populations are likely to emphasize on-farm welfare, the *Labor* variable, which is defined as the proportion of the economically active population in agriculture, is also expected to reflect the practical significance of an agricultural development objective. Similarly, to evaluate the relevance of an energy independence objective, we included *Eind*, a variable that measures the net energy imports as a percentage of domestic consumption. Additionally, we added *Pln*, a variable that measures the proportion of $CO_2$ emissions from liquid fuel to examine whether countries with higher emissions are more likely to adopt renewable energy policies. Finally, we used three measures of comparative advantage, *CA*, to determine whether the biofuel policies are consistent with the trade theory. After these revisions, the empirical model can be expressed as

$$Bpol = a + b_1 Land + b_2 Labor + b_3 Tech + b_4 GDP + b_5 GDP^2 + b_6 Eind + b_7 Pln + b_8 CA + \varepsilon \quad (2)$$

where *Bpol* is a dummy variable indicating whether the country has an ethanol–gasoline mix requirement or a policy considering specific gasoline standard targets at time *t*, *b*s are unknown parameters to be estimated, and $\varepsilon$ is a random error term. All variables are measured over time (1991–2011) for each of the four countries included in the study, but the time and state subscripts are suppressed to simplify the expression.

### 3.2. Data Sources and Model Variables

Most of the data used in this study were obtained from the Food and Agriculture Organization (faostat3.fao.org) and the World Bank (databank.worldbank.org) online databases. However, numerous policy documents and other literature were used to construct the biofuel policy adoption variable [9,50,60,61].

The three comparative advantage indices used in this study—i.e., relative feedstock price, comparative export performance, and revealed comparative advantage—are defined as follows. Since sugarcane is the primary biofuel feedstock used in the LAC4 countries, we constructed the comparative advantage indices based on sugarcane prices (the domestic and international price of

sugar, raw centrifugal) and trade data. The first index, the relative price ratio of raw centrifugal sugar, is defined as the domestic price/international price of sugar, raw centrifugal. The second index, comparative export performance (CEP), was initially proposed by Balassa and subsequently revised and applied by numerous researchers [62,63]. The CEP index is defined as

$$CEP = (X_B^i / X_B) \, / \, (X_w^i / X_w) \tag{3}$$

where, $X_B^i$ is country $B$'s export of good $i$, $X_B$ represents country $B$'s total exports, $X_w^i$ is total world export of good $i$, and $X_w$ measures overall world export.

The third index, revealed comparative advantage ($RCA$), is as defined by Vollrath (1991) and applied in subsequent studies [36,64,65].

$$RCA_a^i = RXA_a^i - RMA_a^i, \text{ where} \tag{4}$$

$$RXA_a^i = (X_a^i / X_n^i) / (X_a^r / X_n^r) \tag{5}$$

$$RMA_a^i = (M_a^i / M_n^i) / (M_a^r / M_n^r) \tag{6}$$

The revealed comparative advantage index (RCA) is the difference between two indices—i.e., relative export advantage ($RXA$, Equation (5)) and relative import advantage ($RMA$, Equation (6)) indices. The superscript $i$ refers to country $i$ and $r$ refer to world minus country $i$ difference in exports (or imports). The subscript $n$ refers to all traded commodities from country $i$ (or world) minus commodity $a$ (sugar, raw centrifugal in this case).

### 3.3. Expected Relationships

First-generation renewable energy products, such as ethanol, are derived from food crops such as corn, sugar beets, and sugarcane as the feedstocks to produce them. Therefore, land and labor are two of the most critical resources in producing these feedstocks. The per capita arable land and economically active population in agriculture are used as proxies for land and labor variables, respectively. Consistent with the literature, the relationship between land and labor variables with biofuel policy adoption decision is expected to be positive [9].

Access to biofuel production technology, technological knowledge, and average income also play a crucial role in biofuel production. Since it is challenging to measure access to technology and technical expertise in the general populace, proxies such as total research and development expenditure (R&D), often expressed as the percentage of total GDP, are used in the literature. However, the use of R&D with GDP in the same regression model creates a multicollinearity problem because they are highly correlated [9]. We use the proportion of the population using the internet as a proxy for technological know-how or access to technology (*Tech*). Since access to production technology and the ability to manage it are essential, the relationship between the *Tech* variable and biofuel policy adoption is expected to be positive.

Furthermore, recent studies show a very close relationship between per capita gross domestic product (*GDP*) and the demand for environmental quality [59,66–68]. In particular, most reviews find an inverted U-shaped relationship between income and environmental pollution [59,66]. This phenomenon is known as the environmental Kuznets curve [59,66]. To account for the potential impact of income on biofuel policy adoption, we include per capita *GDP* as an explanatory variable. The relationship between *GDP* and renewable energy production is expected to be curvilinear (i.e., the coefficient of *GDP* variable is expected to be positive, and the *GDP*$^2$ variable negative).

One of the commonly cited policy objectives for developing biofuel policy is the reduction of dependence on foreign oil (*Eind*). It would be more critical for countries that are heavily dependent on fossil fuel imports to satisfy their energy needs. Fossil fuel import data are often used to measure the extent of dependence on imported oil. Consistent with the literature, this study uses net energy

imports as a measure of the relationship between energy dependence and biofuel policy adoption that is expected to be positive.

Additionally, recent studies have observed that rising $CO_2$ emissions from fossil fuels are one of the primary drivers for developing and adopting sustainable biofuel policies [12,66]. The proportion of $CO_2$ emissions from liquid fuel consumption is used as a proxy to examine the empirical validity of this observation. However, it is not clear whether the relationship between the level of $CO_2$ emission from liquid fuel consumption and biofuel policy adoption would be positive or negative.

Three individual models, each using only one of the comparative advantage measures and the other potential biofuel policy drivers discussed above, are estimated. We expect the relationship between the relative biofuel feedstock price (domestic price/international price) and biofuel adoption decision to be negative because the higher price ratio would imply a shortage in feedstock supply and would discourage the production of renewable fuels. On the other hand, comparative export performance and revealed comparative advantage variables are expected to have a positive relationship with the biofuel policy adoption decision.

## 4. Results and Discussion

### Empirical Results

The summary statistics for the variables included in the models are reported in Table 1. The statistics show that there is substantial variation within as well as across the countries included in the study. For instance, access to the internet during the sample period (1991–2011) ranges from zero percent to fifty-one percent. Similarly, the per capita gross domestic product ranges from about $1200 to about $14,000. As expected, similar patterns are observed in other variables.

**Table 1.** Summary statistics of the variables included in the model.

| Variable | Mean | Std. Dev. | Minimum | Maximum |
|---|---|---|---|---|
| Biofuel Policy Adoption (Binary) | 0.36 | 0.48 | 0.00 | 1.00 |
| Internet Access (% of Population) | 12.10 | 14.05 | 0.00 | 51.00 |
| Per Capita GDP ($'000) | 5.79 | 2.90 | 1.21 | 13.69 |
| Economically Active Population in Agriculture (%) | 7.12 | 2.12 | 3.43 | 10.14 |
| Per Capita Arable Land (Hectares/Capita) | 0.36 | 0.28 | 0.04 | 0.93 |
| Net Energy Imports (% of Energy Use) | −54.94 | 75.35 | −281.19 | 30.44 |
| $CO_2$ Emissions from Liquid Fuel (% of Total) | 2.31 | 1.39 | 0.55 | 4.72 |
| Feedstock Price (Domestic/International) | 1.34 | 0.48 | 0.58 | 3.63 |
| Comparative Export Performance (CEP) | 0.34 | 0.48 | −0.25 | 2.47 |
| Revealed Comparative Advantage (RCA) | 0.36 | 0.36 | 0.00 | 1.50 |

Source: The summary statistics are the author's calculations based on the data obtained from different sources, as described in the data section.

The three binomial probit model estimates are presented in Table 2. The three models are identical except for the variable that measures comparative advantage. The first model (Model 1) uses the relative sugarcane price as the comparative advantage index. The comparative export performance index is used in the second model (Model 2). The third model uses the *revealed comparative advantage* index to measure the comparative advantage in feedstock production (Model 3). The pseudo $R^2$ [9] ranges from 0.56 (Model 1) to 0.73 (Model 3), and the Wald statistics are statistically significant, indicating that all three models fit well with the data. Moreover, all estimated parameters are statistically significant at the five percent level (except in Model 1) and carry the expected signs.

**Table 2.** Probit regression results on biofuel policy adoption.

| Variables | Model 1 | | Model 2 | | Model 3 | |
|---|---|---|---|---|---|---|
| | Coefficient | z-Value | Coefficient | z-Value | Coefficient | z-Value |
| Relative Feedstock Price (RFP) | −2.142 * | −1.99 | | | | |
| Comparative Export Performance (CEP) | | | 9.109 ** | 4.52 | | |
| Revealed Comparative Advantage (RCA) | | | | | 7.812 ** | 4.14 |
| Internet Use (*Tech*) | 0.187 ** | 4.76 | 0.367 ** | 4.22 | 0.325 ** | 4.52 |
| Proportion of Ag. Population (*Labor*) | 0.465 | 1.23 | 3.209 ** | 3.52 | 2.965 ** | 3.62 |
| Per Capita Arable Land (*Land*) | −3.420 | −1.16 | 8.257 * | 2.16 | 7.353 * | 1.87 |
| Net Energy Imports (*Indep*) | 0.030 ** | 4.03 | 0.018 ** | 2.65 | 0.022 ** | 2.88 |
| Per Capita GDP (*Edev*) | 0.425 | 1.09 | 1.319 ** | 2.74 | 1.242 ** | 2.44 |
| Per Capita GDP Squared (*Edev2*) | −0.059 * | −2.33 | −0.109 ** | −2.83 | −0.105 ** | −2.83 |
| $CO_2$ Emissions from Liquid Fuel (*Poll*) | −1.027 ** | −2.55 | −1.272 ** | −2.96 | −1.296 ** | −2.72 |
| Constant | 1.253 | 0.28 | −34.199 ** | −3.45 | −29.967 ** | −3.35 |
| Pseudo $R^2$ | 0.56 | | 0.72 | | 0.73 | |
| Wald | 35.36 | | 29.95 | | 30.91 | |

*, ** Denote statistical significance at the five and one percent level, respectively.

All three variables used to measure the access to production resources—land, labor, and technology—are statistically significant and carry the expected signs (except in Model 1). In particular, the coefficient of the land variable is significant in Model 2 and 3 and holds the expected sign. Moreover, in terms of marginal impact, the land has the second-highest effect (1.79) on biofuel policy adoption decisions (see Table 3). These results are consistent with previous studies [9].

**Table 3.** Marginal effects of comparative advantage on biofuel policy adoption.

| Variable | Model 1 | | Model 2 | | Model 3 | |
|---|---|---|---|---|---|---|
| | Coefficient | z−Value | Coefficient | z−Value | Coefficient | z−Value |
| Relative Feedstock Price | −0.379 ** | −2.71 | | | | |
| Comparative Export Performance | | | 1.272 * | 1.72 | | |
| Revealed Comparative Advantage | | | | | 1.898 ** | 2.82 |
| Internet Use | 0.033 ** | 2.70 | 0.051 * | 1.80 | 0.079 ** | 3.55 |
| Proportion of Agricultural Population | 0.082 | 1.33 | 0.448 * | 2.00 | 0.720 ** | 3.75 |
| Per Capita Arable Land (ha.) | −0.605 | −0.97 | 1.153 * | 1.82 | 1.787 ** | 2.31 |
| Net Energy Imports (% of Use) | 0.005 * | 2.18 | 0.003 | 1.27 | 0.005 * | 1.87 |
| Per Capita GDP ($) | 0.075 | 0.98 | 0.184 | 1.27 | 0.302 * | 1.72 |
| Per Capita GDP Squared ($) | −0.010 * | −1.74 | −0.015 | −1.31 | −0.025 * | −2.01 |
| $CO_2$ Emissions from Liquid Fuel | −0.182 * | −1.89 | −0.178 | −1.29 | −0.315 * | −1.77 |

*, ** Denote statistical significance at the five and one percent level, respectively.

As discussed in the last paragraph, the coefficients of the labor variable are also significant in Model 2 and 3 and carry the expected signs. Moreover, the marginal impact of *Labor* is the third highest (0.72) among the seven explanatory variables included in the model (see Table 3). These results show that countries with a proportionally higher agricultural population are more likely to adopt biofuel policies. In addition to serving as a factor of production, the labor variable (as defined in this study) may be a subject of government policy, mainly when a large proportion of the economically active

population is involved in farming. In particular, a higher percentage of the labor force engaged in agriculture is likely to encourage governments to develop and implement programs, such as renewable energy initiatives, that are likely to enhance farm income. In this sense, these results are also consistent with the rural or agricultural development objectives mentioned in most biofuel policy documents.

Two separate proxies were used to measure the impact of the *Tech* variable on biofuel policy adoption—total research and development (*R&D*) expenditure and the proportion of the population using the internet [9]. As expected, both *Tech* measures have a significantly positive impact on renewable fuel policy, but only the results from the model with internet use variables are presented in Table 2.

Energy independence is another objective mentioned in many policy documents [25,56]. The net energy imports (*Eind*) variable was included in the model to examine whether countries that are highly dependent on imported oil are more likely to adopt biofuel policies. The estimated coefficient of this variable is statistically significant in two models (Model 1 and 3) and carries a positive sign. Thus, as expected, these results show that countries with a higher proportion of imported energy in their domestic consumption are more likely to adopt biofuel policies. It implies that energy independence was one of the primary drivers for adopting biofuel policies in LAC4 countries.

As expected, the relationship between income and renewable energy adoption decision is curvilinear (i.e., positive for *GDP* and negative for $GDP^2$), and both coefficients are statistically significant. Consistent with the previous studies, these results imply that as income increases, the demand for environmental quality increases, and countries are likely to adopt policies such as renewable energy initiatives that help protect the ecosystem and enhance environmental quality.

$CO_2$ is one of the major pollutants emitted from fossil fuel-based transportation systems. Since pollution control is one of the primary objectives of most biofuel policies, we include this variable to control for the behavior of the heavy $CO_2$ emitters. The estimated coefficient of this variable is negative and highly significant, implying that countries with higher $CO_2$ emissions from liquid fuel are less likely to adopt the biofuel policies.

All three proxies used to measure the impact of comparative advantage on biofuel policy adoption decisions have the expected signs (a negative sign for relative feedstock price and positive for revealed comparative advantage and comparative export performance index) and are highly significant. In all three models, the marginal impact of the revealed comparative advantage index is highest (see Table 3). These results imply that comparative advantage primarily drove the biofuel policy adoption decisions in the LAC4 countries. Thus, these policies are consistent with theoretical international trade relationships.

The empirical results show that the probability of a country adopting biofuel policies rises as per capita gross domestic product (income), per capita arable land (land), access to modern technology (proportion of population using the internet), and availability of farm labor (the percentage of economically active population in agriculture) increase. Likewise, countries that are heavily dependent on imported oil are more likely to adopt biofuel policies. The results also show that countries with a higher proportion of $CO_2$ emissions from liquid fuels are less likely to adopt biofuel policies. Moreover, as postulated by international trade theories, comparative advantage in feedstock production played a crucial role in biofuel policy development and adoption in LAC4 countries.

## 5. Conclusions

We examined whether the recent surge in biofuel policy adoption is consistent with international trade theory (comparative advantage), domestic production capacity, and other stated policy goals using panel data from four major LAC countries. Three different indices (i.e., relative feedstock price, comparative export performance, and revealed comparative advantage) are used as proxies for measuring the impact of comparative advantage on biofuel policy adoption decisions. Moreover, most renewable energy initiatives identify greenhouse gas reduction, energy independence, rural development, and sustainability as other desired policy goals.

As expected, empirical results show that countries with a higher dependence on imported oil, a higher proportion of agricultural labor force, greater access to arable land, and higher per capita income are more likely to adopt biofuel policies. On the other hand, countries with a higher level of $CO_2$ emissions from liquid fuels are less likely to adopt such policies. Moreover, the probability of biofuel policy adoption rises as access to modern production and information technologies, such as the internet, increases.

All three indices used to measure the impact of comparative advantage on biofuel policy adoption decisions carry the expected signs and are highly significant, implying that recent policy initiatives to promote production and utilization of renewable fuels in LAC4 countries are consistent with economic theory. Moreover, these results also show that biofuel policy can be used to achieve multiple objectives, including economic development and environmental sustainability.

The managerial and policy implications of this study are that most biofuel policies are consistent with both environmental as well as economic sustainability. Thus, consistent with the recent studies [9,12,69], well-conceived and adequately implemented renewable energy initiatives provide opportunities for enhancing the welfare of all stakeholders involved in producing and consuming biofuel energy products in the long run. Therefore, future studies should consider the importance of comparative advantage and other socioeconomic variables in evaluating the effectiveness of different biofuel policies.

One of the significant limitations of this study is the lack of consistent time-series data on different aspects of biofuel policy development, implementation, production, prices, and utilization. For instance, because of data limitations, we used a binary policy variable that indicates whether a country has implemented a biofuel policy in a particular year, which limits the choice and scope of empirical analysis. Moreover, the quality of data is questionable for most of the countries studied. As more reliable time-series data become available, future studies may utilize sophisticated parametric techniques, such as panel cointegration, or other nonparametric tools to analyze the long-term implications of different biofuel policy alternatives.

**Author Contributions:** Conceptualization, R.N.A. and R.P.-P.; data, R.P.-P.; software, secondary data collection, and analysis, R.N.A.; writing—original draft preparation, R.N.A.; review and editing, R.N.A. and R.P.-P.; funding acquisition, R.N.A. All authors have read and agreed to the published version of the manuscript.

**Funding:** This research was supported partly by the New Mexico Agricultural Experiment Station and partly by USDA Hatch funding.

**Conflicts of Interest:** The authors declare no conflict of interest. The funders had no role in the design of the study; in the collection, analyses, or interpretation of data; in the writing of the manuscript, or in the decision to publish the results.

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
