# Peer review of "Role of Comparative Advantage in Biofuel Policy Adoption in Latin America"

_sustainability, doi:10.3390/su12041411_

Round 1
Reviewer 1 Report
Thank you for the opportunity to review this article. The paper tries to evaluate whether the renewable energy initiatives recently developed and implemented in Latin American and Caribbean countries are consistent with their national resources, policy goals, and the general postulates of economic theory. Main notes are below:
Introduction should be clearly stated research questions and targets first. Also the novelty and input of study in the field should be stressed in the introduction. I would recommend to indicate the importance of this contribution to sustainability research area. Literature review in the field is missing. I would recommend to add a literature review to justify research methodology. Please answer a few questions: what has been done in the field; what kind of methods were used to analyse similar research questions, what results were obtained? It would be logical and interesting to the readers, if the authors add primary statistics (not only the summary) of LAC4 countries. It would be logical to add justification of variables in the section 2. I would recommend to change section 4 title to "Conclusions" Citation requirements (line 73-74; line 112; line 169 etc.). Line 173 references are missing "...recent studies show...". Line 174 references are missing "...most reviews find...". Line 186 references are missing "...recent studies have observed...". Finally, please check carefully the manuscript because minor mistakes are shown (for example: keywords should be written in lowercase; tables should be adjusted based on requirements of the journal; a complete concept with abbreviation is only presented when it is mentioned for the first time…).Author Response
General Comment: Thank you for the opportunity to review this article. The paper tries to evaluate whether the renewable energy initiatives recently developed and implemented in Latin American and Caribbean countries are consistent with their national resources, policy goals, and the general postulates of economic theory. Main notes are below:
We appreciate encouraging comments. Thank you.
Introduction should be clearly stated research questions and targets first. Also the novelty and input of study in the field should be stressed in the introduction. I would recommend to indicate the importance of this contribution to sustainability research area.In addition to adding more reencent literature in several places, we have added the following text (line 75-79):
“In this light, the current study contributes to the existing literature by evaluating the impact of three different comparative advantage measures (i.e., relative feedstock price, relative export performance, and revealed comparative advantage) along with other crucial policy variables including technology, rural development, energy independence, and pollution control on biofuel policy adoption in LAC4 countries.”
Literature review in the field is missing. I would recommend to add a literature review to justify research methodology. Please answer a few questions: what has been done in the field; what kind of methods were used to analyse similar research questions, what results were obtained? It would be logical and interesting to the readers, if the authors add primary statistics (not only the summary) of LAC4 countries. It would be logical to add justification of variables in the section 2.
Although we do not have a separate section for literature review, current literature is paraphrased and cited extensively in the introduction and methodology sections. Moreover, the methodology section includes the following statement at the beginning and several references are mentioned throughout the section including a few new references.
“Most empirical studies use limited dependent variable models such as logit or probit regressions to study new technology adoption decisions [40-43]. Consistent with this literature, a simple conceptual model of biofuel policy adoption decision can be specified as:
(1) .”
I would recommend to change section 4 title to "Conclusions"
Revised as suggested.
Citation requirements (line 73-74; line 112; line 169 etc.). Line 173 references are missing "...recent studies show...". Line 174 references are missing "...most reviews find...". Line 186 references are missing "...recent studies have observed...". Finally, please check carefully the manuscript because minor mistakes are shown (for example: keywords should be written in lowercase; tables should be adjusted based on requirements of the journal; a complete concept with abbreviation is only presented when it is mentioned for the first time…).
New citations are added in all places as recommended and keywords are changed to lower case. However, I am waiting until I receive comments from the copy editor to make changes in table format.
Reviewer 2 Report
In the manuscript, the authors analyzed ethanol policy drivers, using panel data from Argentina, Brazil, Colombia, and Mexico, to evaluate whether the renewable energy initiatives recently developed and implemented in Latin American and Caribbean countries are consistent with their national resource endowments, policy goals, and the general postulates of economic theory.
The topic of the paper is interesting as well as the academic contribution of the work, but the authors should improve their work according to the following indications.
In the introduction: the authors should discuss international situation, regulations, and approaches, and should motivate their research to be of high interest for the addressees; the authors should explain how the article has been structured by presenting the different sections. Literature review should be extended. The authors should discuss how the results can be interpreted in perspective of previous studies and of the working hypotheses. Limitations of the study and future research directions should be addressed. Tables should report the sources. Moderate English changes required.Author Response
General Comments: In the manuscript, the authors analyzed ethanol policy drivers, using panel data from Argentina, Brazil, Colombia, and Mexico, to evaluate whether the renewable energy initiatives recently developed and implemented in Latin American and Caribbean countries are consistent with their national resource endowments, policy goals, and the general postulates of economic theory. The topic of the paper is interesting as well as the academic contribution of the work, but the authors should improve their work according to the following indications.
We appreciate encouraging comments
In the introduction: the authors should discuss international situation, regulations, and approaches, and should motivate their research to be of high interest for the addressees;Since several studies including those published in the Sustainability journal, have provided an overview of the international biofuel policy situation, initially, we decided not to include it in this study. Although we are open to this suggestion, it would hard for us to provide a comprehensive review in a short time frame, mainly because the corresponding author is traveling for a while with limited access to the internet.
the authors should explain how the article has been structured by presenting the different sections. Literature review should be extended. The authors should discuss how the results can be interpreted in perspective of previous studies and of the working hypotheses.
Several new studies have been paraphrased and cited in numerous places in response to other review comments. As recommended we have added the following text in the revised version outlining information included in different sections:
“The remaining segments of the paper are organized as follows: section two presents the conceptual framework and estimating models based on existing literature, defines model variables, and provides data sources, and describes expected relationships between model variables. The third section presents and interprets the empirical results and the last section draws conclusions and explains some of the limitations of the study.”
Limitations of the study and future research directions should be addressed. Tables should report the sources. Moderate English changes required.We have made a few grammatical changes throughout the paper and added the following text describing the limitations of the study:
“One of the significant limitations of this study is the lack of consistent time series data on different aspects of biofuel policy development, implementation, production, prices, and utilization. For instance, because of data limitations, we used a binary policy variable that indicates whether a country has implemented a biofuel policy in a particular year, which limits the choice and scope of empirical analysis. Moreover, the quality of data is questionable for most of the countries. As more reliable time-series data become available, future studies may utilize sophisticated parametric techniques such as panel cointegration or other nonparametric tools to analyze long-term implications of different policy alternatives.”
Round 2
Reviewer 1 Report
Authors have improved the paper according to my observations.
Author Response
No comments to address
Reviewer 2 Report
The authors have not discussed international situation, regulations, and approaches, have not motivate their research to be of high interest for the addressees, have not extended literature review, have not discuss how the results can be interpreted in perspective of previous studies and of the working hypotheses, have not addressed future research directions.
All this is very important in a scientific paper!
Author Response
We appreciate constructive comments from this reviewer and have made an utmost effort to address all of them.
Comment: The authors have not discussed international situation, regulations, and approaches, have not motivate their research to be of high interest for the addressees, have not extended literature review, have not discuss how the results can be interpreted in perspective of previous studies and of the working hypotheses, have not addressed future research directions.
Response: We have added a whole new section in the revised manuscript summarizing the current situation on global biofuel policy adoption, major policy tools used for implementing the policy, and different types of feedstock used for producing bioethanol and biodisel (line# 96 to 189).
We have also revised the introduction section by adding information from new literature and emphasizing how our study adds value to the existing literature. We have also provided future research direction and emphasized the importance of including relative advantage in future studies.
Round 3
Reviewer 2 Report
Now the authors have taken into consideration all the suggestions of my previous review reports, significantly improving their manuscript.